# Postnatal cytomegalovirus infection and bronchopulmonary dysplasia in very low birth weight infants: Influence of diagnostic criteria and viral load in a propensity score–matched cohort study

**Zhihui Qiao**[1,2], **Yingying Bao**[1,2], **Jiajun Zhu**[1,2]*

**1** Department of Neonatology, Women's Hospital, School of Medicine, Zhejiang University, Hangzhou, China, **2** Zhejiang Key Laboratory of Maternal and Infant Health, Zhejiang University, Hangzhou, China

* jiajunzhu@zju.edu.cn

## Abstract

### Objective

To determine the association between postnatal cytomegalovirus (pCMV) infection and bronchopulmonary dysplasia (BPD) under three different criteria as well as the impact of viral load on clinical outcomes in very low birth weight (VLBW) infants.

### Methods

In this retrospective cohort study, the data for VLBW infants with pCMV infection were collected from de-identified medical records and matched 1:1 with non-infected controls using propensity score matching between January 1, 2014, and December 31, 2024, in a tertiary neonatal intensive care unit in China. The primary outcome was the association between pCMV and BPD according to the 2001 National Institute of Child Health and Human Development (NICHD), 2018 NICHD, and 2019 Neonatal Research Network (NRN) criteria. The secondary outcome was healthcare resource utilization stratified by viral load.

### Results

Seventy-four infants with pCMV infection were matched to 74 infants without pCMV infection. After the adjustment for confounders, pCMV infection was not significantly associated with BPD defined by the 2001 NICHD criteria (adjusted odds ratio [aOR], 8.26; 95% confidence interval [CI], 2.14–31.85; p < 0.01). However, it was associated with BPD under both the 2018 NICHD and 2019 NRN criteria (aOR 11.05, 95% CI, 4.00–30.52; p < 0.001 for both). Moreover, pCMV was significantly associated with higher severity grades (moderate to severe 2001 NICHD: aOR, 10.32; Grade II–III per the 2018 NICHD: aOR, 11.47; Grades 2–3 per the 2019 NRN: aOR, 12.81; all

**Data availability statement:** All relevant data are within the manuscript and its Supporting Information files.

**Funding:** This study was funded by the National Key Research and Development Program of China (Grant No. 2022YFC2703500) and the 4+X Clinical Research Project of Women's Hospital, School of Medicine, Zhejiang University (Grant No. ZDFY2021-4X205). The funders had no role in study design, data collection and analysis, decision to publish, or preparation of the manuscript.

**Competing interests:** The authors have declared that no competing interests exist.

$p < 0.001$). Infants with high viral loads ($\geq 1.34 \times 10^5$ copies/mL) require significantly more healthcare resources than infants with low viral loads (all $p = 0.001$).

## Conclusions

The association between pCMV infection and BPD was influenced by the diagnostic criteria. A higher cytomegalovirus load in VLBW infants was associated with more severe respiratory morbidity and greater healthcare utilization.

## Introduction

Bronchopulmonary dysplasia (BPD) is the most common complication of preterm infants [1] and among the most important causes of adverse long-term outcomes of preterm infants [2]. The diagnostic criteria for BPD have been updated continuously since they were first proposed in 1967 [3]. The definition of BPD based on severity grading, published in 2001 by the National Institute of Child Health and Human Development (NICHD), has been widely used [4]. The NICHD revised the BPD diagnostic criteria in 2018, proposing new graded diagnostic criteria (hereafter 2018 NICHD criteria) [5]. In 2019, the Neonatal Research Network (NRN) proposed definitions for predicting respiratory and neurodevelopmental dysplasia outcomes at a corrected age of 18–26 months based on evidence-based medicine (hereafter 2019 NRN criteria) (Table 1) [6]. Previous studies reported that different diagnostic criteria for BPD have different diagnostic and prognostic predictive abilities [7].

Cytomegalovirus (CMV) is a common cause of perinatal infections. Congenital CMV (cCMV) infection is diagnosed via polymerase chain reaction (PCR) or viral culture of urine samples, respiratory tract secretions, blood, or cerebrospinal fluid obtained within the first 3 weeks of life. Postnatal cytomegalovirus (pCMV) infection is defined as a cytomegalovirus infection after 3 weeks of age upon the exclusion of cCMV infection [8]. Possible routes of pCMV infection in infants include breast milk, blood products, and horizontal transmission. In the current era, with the reduction in CMV transmission via blood products achieved through seronegative donor selection and leukoreduction, exposure to breast milk from CMV-seropositive mothers has emerged as the most common transmission route [9,10]. Moreover, pCMV is usually asymptomatic in term infants but often symptomatic in preterm infants, especially those who are very preterm (gestational age [GA] < 32 weeks) or have a very low birth weight (VLBW; birth weight [BW] < 1500 g), leading to symptomatic manifestations including sepsis, pneumonia, thrombocytopenia, neutropenia, hepatitis, colitis, and occasionally death [11,12]. Notably, pCMV infection is a modifiable risk factor that may exacerbate BPD development and severity.

The pathogenesis of BPD is complex and involves a variety of factors such as oxygen toxicity, ventilator-induced lung injury, inflammation, and impaired pulmonary vascular development [13–15]. In fact, pCMV infection may exacerbate BPD development and affect long-term clinical outcomes [16,17]. Several studies have suggested

**Table 1. Different definitions of BPD.**

| 2001 NICHD | Mild | Room air | | | |
|---|---|---|---|---|---|
| | Moderate | $FiO_2 < 30$ | | | |
| | Severe | $FiO_2 \geq 30$ and/or NCPAP and/or IMV | | | |
| 2018 NICHD | | IMV[a] | NCPAP, NIPPV, NC ≥ 3 L/min | NC ≥ 1 to < 3 L/min, hood $O_2$ | NC < 1 L/min |
| | Grade I | – | 21 | 22-29 | 22-70 |
| | Grade II | 21 | 22-29 | ≥30 | > 70 |
| | Grade III | >21 | ≥30 | – | – |
| | Grade IIIa | Death from respiratory causes between 14 days after birth and 36 weeks' PMA | | | |
| 2019 NRN | Grade 1 | NC ≤ 2 L/min | | | |
| | Grade 2 | NC > 2 L/min or non-invasive positive airway pressure | | | |
| | Grade 3 | IMV | | | |

CPAP, continuous positive airway pressure; $FiO_2$, fraction of inspired oxygen; IMV, invasive mechanical ventilation; NC, nasal cannula; NCPAP, nasal continuous positive airway pressure; NICHD, National Institute of Child Health and Human Development; NIPPV, non-invasive positive pressure ventilation; NRN, Neonatal Research Network; PMA, postmenstrual age

[a]Excluding infants ventilated for primary airway disease or central respiratory control conditions. Values are percentages.

Severity grades are presented using the nomenclature specific to each criterion: mild, moderate, and severe as per the 2001 NICHD definition; Grades I, II, and III as per the 2018 NICHD definition; and Grades 1, 2, and 3 as per the 2019 NRN definition.

that CMV infection may exacerbate BPD progression by affecting lung development and triggering oxidative stress and inflammatory responses [18–20].

However, original studies, reviews, and meta-analyses report inconsistent conclusions regarding the association between pCMV infection and BPD, which may be partially due to varying definitions of BPD [21–24]. Therefore, this study aimed to explore the association between pCMV infection and BPD using three different criteria within a retrospective cohort of VLBW infants and explore whether this association was influenced by CMV viral load.

## Materials and methods

### Study population

This retrospective cohort study was conducted at a tertiary NICU of the Women#39;s Hospital, School of Medicine, Zhejiang University, from January 1, 2014, to December 31, 2024. This NICU admits approximately 500 preterm infants with a GA < 32 weeks per annum. The study included the data of neonates admitted with a GA < 32 weeks and/or BW < 1500 g collected from de-identified medical records. To rule out cCMV, all preterm infants were routinely screened via urine and respiratory secretion samples within the first 48–72 h and repeated 2–3 weeks after birth. A pCMV infection was diagnosed when testing was prompted by clinical suspicion beyond 3 weeks of age based on relevant symptoms (e.g., sepsis-like syndrome, gastrointestinal manifestations) or laboratory abnormalities (e.g., thrombocytopenia, transaminitis). The non-pCMV group included infants who tested negative for CMV after 3 weeks or did not undergo testing in the absence of clinical indications. Infants who with a cCMV infection, who died, or who had a hospitalization duration ≤ 21 days were excluded. The preterm infants with pCMV were included, while non-pCMV control infants were matched 1:1 using propensity score–matched (nearest-neighbor method; caliper width, 0.01) based on the year of birth, sex, GA, BW, and small for GA (SGA) status. cCMV was defined as CMV detected by PCR in urine samples and respiratory secretions within the first 21 days of life. pCMV infection was defined as the detection of CMV for more than 21 days and cCMV-negative. pCMV testing was also performed in all infants who died after 21 days of age.

 

### Ethics review

This study was reviewed and approved by the Ethics Committee of Women's Hospital, School of Medicine, Zhejiang University (IRB-20250209-R). Because all patient information was de-identified, the need for assent or consent from eligible patients and their legal guardians was waived. The data were accessed for research purposes from June 10 to July 11, 2025.

### Primary outcomes

This study compared differences in clinical data and disease incidence between the two groups. The primary study objective was to assess the association between pCMV infection and BPD using different definitions.

### Secondary outcomes

Duration of hospitalization, hospitalization cost, duration of mechanical ventilation, and respiratory support were compared between the pCMV and non-pCMV groups by different CMV loads.

### Statistical analysis

The Statistical Package for the Social Sciences (version 24.0; IBM Corp., Armonk, NY, USA) software was used for statistical analysis. Non-normally distributed measurements are expressed as M (Q1, Q3) and were tested using non-parametric tests. Categorical variables are expressed as number of cases and percentages and were tested using the $\chi^2$ test or Fisher's exact test. Multivariable logistic regression models were subsequently conducted that accounted for possible confounding factors: Model 1 (crude); Model 2, which was adjusted for matching variables (GA, BW, sex, SGA status); Model 3, which was additionally adjusted for early clinical variables available prior to the pCMV diagnosis (mode of delivery, premature rupture of membranes, 1- and 5-min Apgar scores, early IMV, surfactant administration, early-onset sepsis); Model 4, which included all variables from Model 3 plus potential mediators or later-occurring clinical factors (late-onset sepsis, red blood cell transfusion, receipt of diuretics, receipt of glucocorticoids); and Model 5, which included all variables from Model 4 plus treatment for pCMV. The receiver operating characteristic (ROC) curve for the presence of BPD was plotted by CMV load. The optimal threshold is calculated by determining the optimal Youden index. Differences in respiratory outcomes between infants with high versus low CMV loads were compared according to the optimal threshold for CMV load threshold. Values of $p < 0.05$ were considered statistically significant.

## Results

Of the 6112 infants with a GA < 32 weeks and/or BW < 1500 g registered in the NICU during this period, 74 with pCMV infection were matched to 74 pCMV-negative infants using propensity scores (Fig 1). Among the 74 pCMV-infected infants, 28 (37.8%) received ganciclovir therapy and 46 (62.2%) did not receive ganciclovir therapy.

The baseline characteristics before versus after propensity score matching are shown in Table 2. Standardized mean differences (SMD) demonstrated the effective matching of key variables. The post-matching diagnostic assessments indicated that the majority of variables were well-balanced. However, a residual imbalance in SGA status remained (SMD = 0.228).

The participants' clinical characteristics and primary outcomes are summarized in Table 3. Compared to controls, infants with pCMV had significantly higher rates of late-onset sepsis and blood transfusions, more exposure to diuretics and corticosteroids, longer respiratory support durations, longer hospitalization durations, and greater costs (all p < 0.05).

The relationship between pCMV infection and BPD is shown in Table 3. Although the overall incidence of BPD as determined by the 2001 NICHD criteria did not differ significantly, the pCMV group exhibited a significantly higher proportion of moderate to severe BPD cases. Conversely, using the 2018 NICHD and 2019 NRN criteria, pCMV infection was

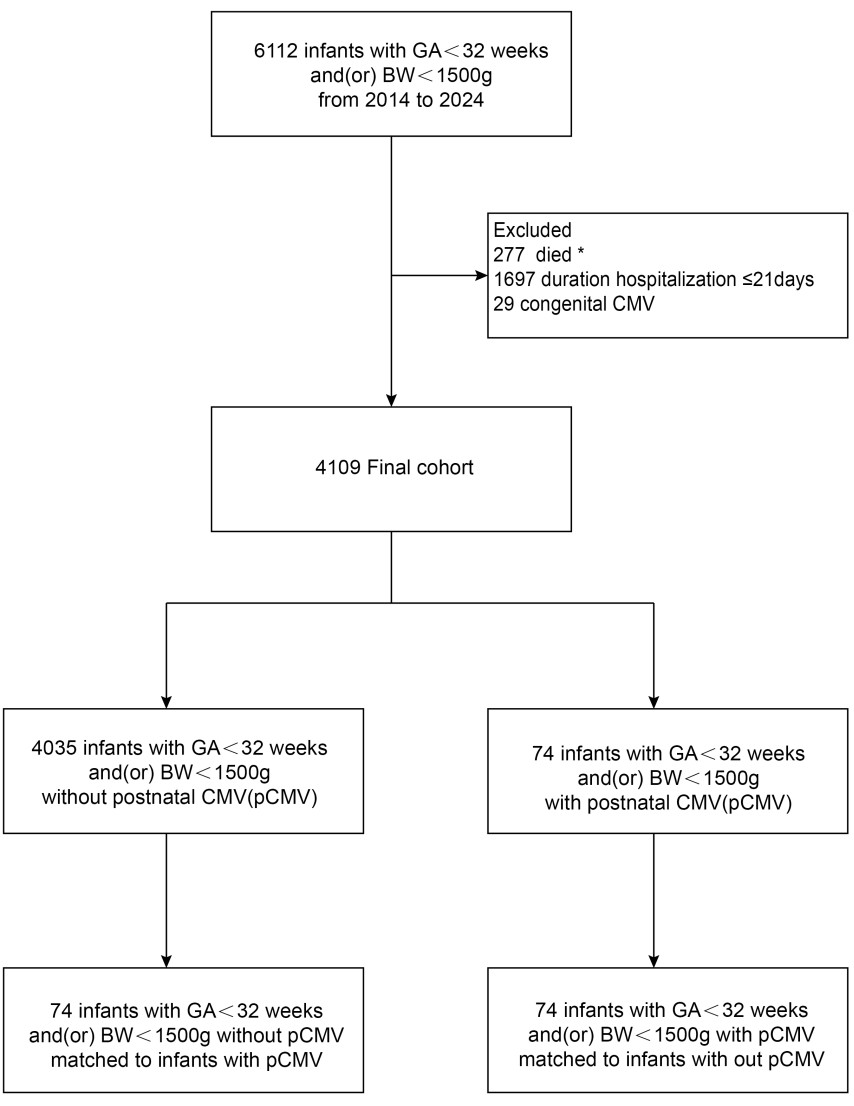

**Fig 1. Flowchart.**

**Table 2. Baseline characteristics before versus after propensity score matching.**

| Variable | Unmatched Cohort (n = 4109) | | | | Propensity Score–Matched Cohort (n = 148) | | | |
|---|---|---|---|---|---|---|---|---|
| | No pCMV(n = 4035) | pCMV(n = 74) | P | SMD | No pCMV(n = 74) | pCMV(n = 74) | P | SMD |
| GA, weeks | 30.00(29.00, 31.00) | 28.00(27.00, 29.00) | <0.001 | 1.118 | 28.00(27.00, 29.00) | 28.00(27.00, 29.00) | 0.579 | 0.090 |
| BW, g | 1345.00(1140.00, 1530.00) | 1060.00(892.50, 1187.50) | <0.001 | 1.302 | 1022.50(870.00, 1150.00) | 1060.00(892.50, 1187.50) | 0.581 | 0.094 |
| Male sex | 2192.00 (54.32) | 41.00 (55.41) | 0.853 | 0.022 | 45.00 (60.81) | 41.00 (55.41) | 0.505 | 0.109 |
| SGA | 372.00 (9.22) | 11.00 (14.86) | 0.098 | 0.159 | 5.00 (6.76) | 11.00 (14.86) | 0.112 | 0.228 |

BW, birth weight; GA, gestational age; pCMV, postnatal CMV; SGA, small for gestational age; SMD, standardized mean difference

Data are shown as median (interquartile range) or n (%), as appropriate.

**Table 3. Clinical characteristics and primary outcomes by study group and BPD definition.**

| Variable | No Postnatal CMV (n=74) | Postnatal CMV (n=74) | P |
|---|---|---|---|
| Premature rupture of membranes | 30.00 (40.54) | 36.00 (48.65) | 0.321 |
| Cesarean delivery | 28.00 (37.84) | 26.00 (35.14) | 0.733 |
| 1-min Apgar score | 9.00 (6.00, 9.00) | 9.00 (6.00, 9.00) | 0.787 |
| 5-min Apgar score | 9.00 (9.00, 10.00) | 9.00 (8.00, 10.00) | 0.610 |
| Intubation in the delivery room | 18.00 (24.32) | 16.00 (21.62) | 0.696 |
| Surfactant administration | 42.00 (56.76) | 47.00 (63.51) | 0.401 |
| Early-onset sepsis | 8.00 (10.81) | 10.00 (13.51) | 0.615 |
| Days of IMV | 0.00 (0.00, 3.25) | 0.00 (0.00, 2.25) | 0.854 |
| Days of respiratory support | 40.00 (14.75, 65.75) | 57.00 (32.50, 84.25) | 0.016 |
| Duration of hospitalization | 68.00 (56.75, 87.25) | 78.00 (64.25, 96.00) | 0.026 |
| Hospitalization cost, CNY | 96725.79 (61651.43, 123797.67) | 113451.00 (80700.25, 150344.50) | 0.008 |
| Pulmonary hemorrhage | 4.00 (5.41) | 6.00 (8.11) | 0.512 |
| NEC | 8.00 (10.81) | 9.00 (12.16) | 0.797 |
| PDA | 46.00 (62.16) | 44.00 (59.46) | 0.736 |
| hsPDA | 35.00 (47.30) | 28.00 (37.84) | 0.245 |
| Grade 3 or 4 IVH | 0.00 (0.00) | 4.00 (5.40) | 0.128 |
| Late-onset sepsis | 29.00 (39.19) | 55.00 (74.32) | 0.000 |
| RBC transfusion required | 42.00 (56.76) | 58.00 (78.38) | 0.005 |
| RBC transfusions | 1.00 (0.00, 3.00) | 2.00 (1.00, 4.00) | 0.003 |
| Received diuretic | 24.00 (32.43) | 36.00 (48.65) | 0.045 |
| Received glucocorticosteroids | 7.00 (9.46) | 19.00 (25.68) | 0.010 |
| Stage per 2001 NICHD BPD criteria | 50.00 (67.57) | 60.00 (81.08) | 0.060 |
| Mild | 28.00 (37.84) | 10.00 (13.51) | 0.001 |
| Moderate to severe | 22.00 (29.73) | 50.00 (67.57) | 0.000 |
| Stage per 2018 NICHD BPD definition | 24.00 (32.43) | 50.00 (67.57) | 0.000 |
| I | 19.00 (25.68) | 25.00 (33.78) | 0.281 |
| II-III | 5.00 (6.76) | 25.00 (33.78) | 0.000 |
| Stage per 2019 NDN BPD criteria | 24.00 (32.43) | 50.00 (67.57) | 0.000 |
| 1 | 17 (22.97) | 19 (25.68) | 0.702 |
| 2-3 | 7 (9.46) | 31 (41.89) | 0.000 |

BPD, bronchopulmonary dysplasia; CNY, Chinese Yuan; hsPDA, hemodynamically significant patent ductus arteriosus; IMV, invasive mechanical ventilation; IVH, intraventricular hemorrhage; NEC, necrotizing enterocolitis; NICHD, National Institute of Child Health and Human Development; NRN, Neonatal Research Network; PDA, patent ductus arteriosus; RBC, red blood cell

Data are shown as median (interquartile range) or n (%), as appropriate.

associated with an increased overall incidence of BPD, with infected infants predominantly classified into grades II–III/2–3 severe categories (Table 3).

To assess the independent relationship between pCMV infection and BPD while accounting for potential confounding and mediation effects, we conducted a series of multivariable logistic regression analyses across five sequential adjustment models. The unadjusted association (Model 1) revealed a significantly elevated risk of BPD as defined by the 2018 NICHD and 2019 NRN criteria. After the adjustment solely for the matching variables (Model 2), the strength of these associations increased substantially. Notably, Model 3 included additional adjustments for early clinical factors available before the pCMV diagnosis, such as delivery mode, premature rupture of membranes, Apgar scores, and early respiratory interventions. After these adjustments, the positive associations between pCMV and the 2018 and 2019 BPD definitions

remained highly significant and were further strengthened (aOR, 11.05; 95% CI, 4.00–30.52). Further adjustments for variables that may arise after pCMV infection and act as potential mediators, including late-onset sepsis, need for transfusions, and administration of diuretics and glucocorticoids in Model 4, or for pCMV treatment in Model 5, did not significantly reduce the effect estimates (Table 4).

ROC curve analysis assessed the diagnostic performance of CMV load for BPD as per the 2018 NICHD criteria, identifying an optimal cut-off at $1.34 \times 10^5$ copies/mL (Fig 2). To evaluate the relationship between disease severity and viral replication extent, we stratified the pCMV cohort based on an optimal viral load threshold of $1.34 \times 10^5$ copies/mL into high (CMV DNA-H, n = 45) and low (CMV DNA-L, n = 29) viral load groups. Compared to the CMV DNA-L group, infants in the CMV DNA-H group exhibited a lower median BW, longer duration of invasive mechanical ventilation (IMV) and total respiratory support, extended hospitalization period, and substantially higher hospitalization costs (all p < 0.05). Although both groups exhibited high BPD rates according to the 2001 NICHD definition, the CMV DNA-H group had a significantly higher proportion of moderate to severe cases (80.00% vs. 48.30%, p = 0.004). With the 2018 NICHD and 2019 NRN criteria, the overall incidence of BPD was nearly twice as high in the CMV DNA-H group (82.22% vs. 44.83%, p = 0.001); this group also had a significantly higher proportion of the most severe grades (Grades II–III/2–3) (Table 5).

## Discussion

This study demonstrated the association between pCMV infection and BPD using three distinct diagnostic criteria in a cohort of VLBW infants spanning an 11-year period in a tertiary Chinese NICU. Notably, after the adjustment for confounders, infants with pCMV had a 11.05-fold increased odds of developing BPD according to the 2018 NICHD and 2019 NRN criteria. The incidence of pCMV infection was 1.8%, a rate that is consistent with that of one previous report [18] but higher than another [16]. Furthermore, pCMV infection, particularly with a high viral load, was associated with worse respiratory outcomes, including prolonged IMV and extended hospitalization durations.

**Table 4. Uni- and multivariate logistic regression analyses of pCMV and BPD by various definitions.**

| Outcome | Model 1 | | Model 2 | | Model 3 | | Model 4 | | Model 5 | |
|---|---|---|---|---|---|---|---|---|---|---|
| | OR (95% CI) | P | aOR (95% CI) | P | aOR (95% CI) | P | aOR (95% CI) | P | aOR (95% CI) | P |
| Stage per 2001 NICHD BPD criteria | 2.06 (0.96-4.39) | 0.06 | 5.42 (1.72-17.06) | <0.01 | 8.26 (2.14-31.85) | <0.01 | 9.58 (1.88-48.86) | <0.01 | 7.99 (1.46-43.75) | <0.01 |
| Moderate to severe | 3.68 (1.94-6.99) | <0.001 | 8.74 (3.79-20.14) | <0.001 | 10.32 (4.18-25.49) | <0.001 | 9.84 (3.56-27.26) | <0.001 | 7.59 (2.56-22.56) | <0.001 |
| Stage per 2018 NICHD BPD definition | 4.34 (2.18-8.64) | <0.001 | 8.69 (3.50-21.59) | <0.001 | 11.05 (4.00-30.52) | <0.001 | 10.86 (3.03-38.92) | <0.001 | 7.84 (2.02-30.48) | <0.01 |
| II-III | 4.85 (2.51-9.38) | <0.001 | 8.97 (4.10-19.60) | <0.001 | 11.47 (4.93-26.65) | <0.001 | 10.19 (3.76-27.63) | <0.001 | 9.43 (3.20-27.79) | <0.001 |
| Stage per 2019 NDN BPD criteria | 4.34 (2.18-8.64) | <0.001 | 8.69 (3.50-21.59) | <0.001 | 11.05 (4.00-30.52) | <0.001 | 10.86 (3.03-38.92) | <0.001 | 7.84 (2.02-30.48) | <0.01 |
| 2-3 | 4.95 (2.57-9.56) | <0.001 | 9.98 (4.47-22.28) | <0.001 | 12.81 (5.38-30.50) | <0.001 | 12.97 (4.65-36.19) | <0.001 | 10.41 (3.45-31.39) | <0.001 |

aOR, adjusted odds ratio; BPD, bronchopulmonary dysplasia; CI, confidence interval; IMV, invasive mechanical ventilation; NICHD, National Institute of Child Health and Human Development; NRN, Neonatal Research Network; OR, odds ratio; RBC, red blood cell

Model 1: Crude

Model 2: Model 1 + gestational age + birth weight + sex + small for gestational age

Model 3: Model 1 + Model 2 + early IMV + surfactant administration + Apgar score + premature rupture of membranes + cesarean delivery + early-onset sepsis

Model 4: Model 1 + Model 2 + Model 3 + RBC transfusion + diuretics + glucocorticosteroids + late-onset sepsis

Model 5: Model 1 + Model 2 + Model 3 + Model 4 + treatment

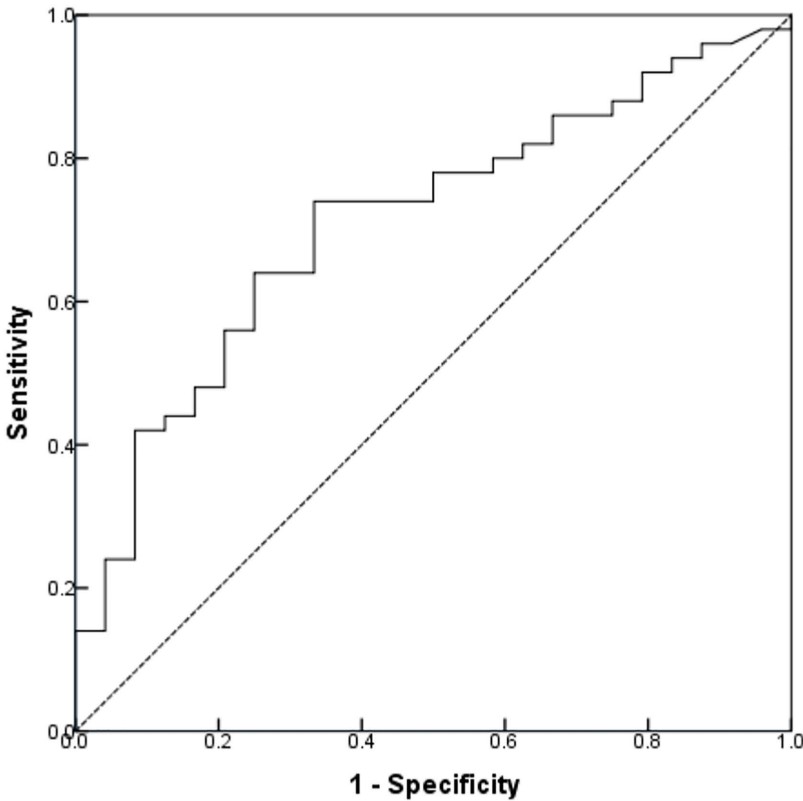

**Fig 2. ROC curve for diagnosing BPD2018 when using CMV loads alone.** The average of sensitivity and specificity is maximized at 74.0% and 66.7%, respectively, with the optimal cut-off CMV organism loads of $1.34 \times 10^5$ copies/mL. The AUC was 0.711.

The enhancement of medical interventions has led to an increase in the survival rate of preterm infants, but concurrently with an increase in the prevalence of BPD [25–27]. As our understanding of the pathophysiology and management of BPD have evolved, the diagnostic criteria have moved from the widely used but less specific 2001 NICHD definition toward more contemporary standards (2018 NICHD and 2019 NRN criteria) that better reflect modern and new noninvasive respiratory support modalities, in particular, the use of heated humidified high-flow nasal cannula, and are more predictive of long-term outcomes [28–30]. Previous studies suggested that the 2018 NICHD and 2019 NRN definitions could reliably and practically predict long-term outcomes [31–34]. In line with this evolution, our findings demonstrate that the stronger association under the 2018/2019 criteria stems from a focus on respiratory support rather than oxygen concentration alone. In addition, the 2018 NICHD standard classifies infants who die of severe lung disease between 14 days after birth and a postmenstrual age of 36 weeks as having grade III (A) BPD, the most severe form. However, no pCMV was detected in any of the infants who died after 21 days; therefore, we did not include any cases of BPD IIIA.

To our knowledge, this is the first study comparing the relationship between pCMV infection and BPD using three distinct diagnostic criteria. Our study found significant differences in the 2018 NICHD and 2019 NRN criteria between pCMV-infected and non-pCMV-infected groups in the overall incidence and Grades II–III/2–3. In contrast, the 2001 NICHD criteria revealed significant differences between the moderate and severe groups only. However, the overall diagnostic rate did not differ significantly between groups. This suggests that pCMV infection may exacerbate BPD severity. However, alternative interpretations must be considered. The observed differences may be attributable, at least in part, to inherent variations in the sensitivity of the diagnostic criteria themselves rather than exclusively indicating a distinct

**Table 5. Respiratory outcomes of CMV DNA-L versus CMV DNA-H groups.**

| Variable | CMV DNA-L (n = 29) | CMV DNA-H (n = 45) | P |
|---|---|---|---|
| Gestational age, weeks | 28 (28, 29) | 28 (26, 29) | 0.095 |
| Birth weight, g | 1170.00 (975.00, 1280.00) | 1000.00 (825.00, 1100.00) | 0.001 |
| Duration hospitalization | 68.00 (55.00, 79.00) | 89.00 (74.00, 99.00) | 0.001 |
| Hospitalization cost, CNY | 86595.00 (66060.00, 101617.00) | 142202.00 (102186.00, 156589.00) | 0.001 |
| Days of IMV | 0.00 (0.00, 0.00) | 0.00 (0.00, 5.00) | 0.022 |
| Days of respiratory support | 38.00 (23.00, 51.00) | 72.00 (51.00, 89.00) | 0.001 |
| Stage per 2001 NICHD BPD definition | 21 (72.41) | 39 (86.67) | 0.126 |
| Mild | 7 (24.14) | 3 (6.67) | 0.072 |
| Moderate to severe | 14(48.30) | 36 (80.00) | 0.004 |
| Stage per 2018 NICHD BPD definition | 13 (44.83) | 37 (82.22) | 0.001 |
| I | 7.00 (24.14) | 18.00 (40.00) | 0.159 |
| II-III | 6.00 (20.70) | 19.00 (42.20) | 0.056 |
| Stage per 2019 NDN BPD definition | 13.00 (44.83) | 37.00 (82.22) | 0.001 |
| 1 | 7.00 (24.10) | 12.00 (26.70) | 0.808 |
| 2-3 | 6.00 (20.70) | 25.00 (55.60) | 0.003 |

BPD, bronchopulmonary dysplasia; CMV DNA-H, high cytomegalovirus viral DNA load group (≥1.34 × 10$^5$ copies/mL); CMV DNA-L, low cytomegalovirus viral DNA load group (<1.34 × 10$^5$ copies/mL); CNY, Chinese Yuan; IMV, invasive mechanical ventilation; NICHD, National Institute of Child Health and Human Development; NRN, Neonatal Research Network

Data are shown as median (interquartile range) or n (%), as appropriate.

pathophysiology specific to pCMV. The 2001 definition, which relies on FiO$_2$ levels assessed at a fixed time point, may misclassify infants on substantial non-invasive support with low oxygen concentrations as merely having "mild" disease. Conversely, the 2018 NICHD and 2019 NRN criteria, which assess severity based on type of respiratory support, are inherently more sensitive at identifying prolonged respiratory insufficiency from any cause. Nevertheless, our data substantiate the significant role of pCMV in driving severe diseases. The robust and independent association between pCMV and the highest disease severity (Grades II–III/2–3) within modern frameworks indicates that pCMV is a key contributor to severe, protracted respiratory morbidity. This finding aligns with that of a recent report by Tran et al. [35], who identified an elevated risk of moderate to severe BPD under the 2001 definition. The 2018 NICHD and 2019 NRN criteria are more stringent and better predictors of long-term adverse pulmonary outcomes [36]. The strong association of pCMV with BPD under these definitions further suggests that pCMV not only exacerbates disease severity but may also contribute to a poorer long-term prognosis.

Our findings indicate that an elevated CMV viral load (≥1.34 × 10$^5$ copies/mL) is associated with an increased incidence of BPD as well as worsened respiratory outcomes, prolonged IMV duration, and extended respiratory support in VLBW infants (Table 5). This aligns with prior reported that a high CMV DNA load (>50,000 copies/mL) was independently associated with organ hemorrhage (OR, 3.61; 95% CI, 1.03–12.65) [37]. Additionally, a higher viral load has been correlated with more severe manifestations and poorer outcomes in cases of CMV pneumonia among immunocompetent hosts and CMV disease in transplant recipients [38,39]. Although specific thresholds have not been established, a higher viral load is associated with symptomatic infection and poor prognosis [40–42]. The potential mechanisms by which a high CMV load exacerbates BPD are multifaceted. First, elevated viremia can trigger a systemic inflammatory response or viral sepsis characterized by the release of proinflammatory cytokines [43]. This systemic inflammation may exacerbate the inflammatory environment in the developing preterm lung, thereby disrupting alveolarization and promoting fibrosis, the key features of BPD pathogenesis. Second, the CMV induces endothelial injury and dysfunction [44]. Within the pulmonary vasculature, this can impair angiogenesis and vascular remodeling, further worsening BPD severity. Finally, human CMV

(HCMV) infection is typically managed by a robust immune response; however, in individuals with compromised immune systems, HCMV can replicate at high levels, leading to end-organ disease (EOD) [45]. In compromised hosts, particularly neonates who are preterm or have a VLBW, the absence of T cell and natural killer (NK) cell function facilitates unrestricted viral replication. This results in high viral loads and ultimately culminates in EOD, which is characterized by direct viral damage, such as retinitis and pneumonia, and indirect effects, including endothelial injury [46,47]. However, prior research predominantly concentrated on cCMV infections; thus, the relationship between viral load and disease severity in pCMV infections requires further exploration.

The primary strength of our study is that it analyzed the relationship between pCMV and BPD based on three definitions, which compensated for the limitations of previous studies that used a single diagnostic criterion and the inclusion of definitions from the 2018 NICHD and 2019 NRN, which can better predict adverse outcomes [48–50]. Another advantage is its long period spanning more than a decade of practice studies and the use of propensity score matching to similar infants for outcome comparisons.

Our study had several limitations. First, its retrospective single-center design and limited sample size may have reduced its statistical power to detect significant differences in outcomes despite being conducted in a large tertiary NICU. Second, an imbalance persisted in the SGA following the matching process (SMD = 0.228), potentially attributable to the somewhat over-constrained simultaneous matching for SGA, GA, and BW, in conjunction with the relatively small size of the exposed cohort (n = 74). Although this imbalance was accounted for in the regression models, residual confounding may still be present. Nevertheless, the effect estimates across various models were consistent, indicating that the impact of such confounding is likely minimal. Third, the diagnosis of pCMV infection was determined using a symptom-driven testing strategy rather than universal screening. Although this method aligns with real-world clinical practice and is prevalent in the literature [23,51], it introduces a potential misclassification bias. Therefore, some asymptomatic pCMV infections may have been overlooked in the control group. This non-differential misclassification is likely to bias our findings toward the null hypothesis by attenuating the intergroup differences. Consequently, the observed associations between pCMV infection and BPD, particularly when newer diagnostic criteria were applied, are likely conservative estimates of the true effect.

In summary, pCMV was associated with an increased incidence of BPD and moderate to severe BPD in VLBW infants according to two recently updated definitions. Further research is needed to develop pCMV prevention and management measures for VLBW infants.

## Supporting information

**S1 File. Participants clinical data.**
(XLSX)

## Acknowledgments

We acknowledged all the medical staffs who were involved in our study. We would like to thank Editage (www.editage.cn) for English language editing.

## Author contributions

**Conceptualization:** jiajun Zhu.

**Data curation:** Zhihui Qiao.

**Formal analysis:** Yingying Bao.

**Funding acquisition:** jiajun Zhu.

**Investigation:** Zhihui Qiao.

**Methodology:** jiajun Zhu.

**Project administration:** Zhihui Qiao, Yingying Bao, jiajun Zhu.

**Resources:** jiajun Zhu.

**Software:** Zhihui Qiao.

**Supervision:** Yingying Bao.

**Validation:** Zhihui Qiao, Yingying Bao.

**Writing – original draft:** Zhihui Qiao, Yingying Bao.

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
