## [Decision Letter · Decision Letter 0]

26 Dec 2025

Dear Dr. Zhu,

Thank you for submitting your manuscript to PLOS ONE. After careful consideration, we feel that it has merit but does not fully meet PLOS ONE’s publication criteria as it currently stands. Therefore, we invite you to submit a revised version of the manuscript that addresses the points raised during the review process.

We look forward to receiving your revised manuscript.

Kind regards,

Kazumichi Fujioka

Academic Editor

PLOS One

Journal Requirements:

https://journals.plos.org/plosone/s/file?id=ba62/PLOSOne_formatting_sample_title_authors_affiliations.pdf ..

“National Key Research and Development

Program of China, Grant/Award Number:

2022YFC2703500; 4+X Clinical Research

Project of Women's Hospital, School of

Medicine, Zhejiang University,

Grant/Award Number: ZDFY2021‐4X205”

4. Please ensure that you include a title page within your main document. You should list all authors and all affiliations as per our author instructions and clearly indicate the corresponding author.

5. Please amend your manuscript to include your abstract after the title page.

6. Please include a separate caption for each figure in your manuscript.

7. Please include your tables as part of your main manuscript and remove the individual files. Please note that supplementary tables (should remain/ be uploaded) as separate "supporting information" files

Reviewers' comments:

Reviewer's Responses to Questions

**Comments to the Author**

1. Is the manuscript technically sound, and do the data support the conclusions?

Reviewer #1: Partly

Reviewer #2: Yes

2. Has the statistical analysis been performed appropriately and rigorously?

Reviewer #1: Yes

Reviewer #2: Yes

3. Have the authors made all data underlying the findings in their manuscript fully available?

Reviewer #1: Yes

Reviewer #2: Yes

4. Is the manuscript presented in an intelligible fashion and written in standard English?

Reviewer #1: No

Reviewer #2: Yes

Reviewer #1: Thank you for the opportunity to review the manuscript entitled “Postnatal cytomegalovirus infection and bronchopulmonary dysplasia in very low birth weight infants: influence of diagnostic criteria and viral load in a propensity score-matched cohort study”. This study addresses an important and clinically relevant issue in neonatal intensive care-the association between pCMV infection and BPD in VLBWIs. However, several major methogological and interpretive concerns should be addressed before its being considered for publication.

1.The manuscript needs careful writing editing. Many sentences contain grammatical mistakes and wrong sentence structure, which make the article difficult to understand.

2.Introduction: Necessary references need to be added to the concept of cCMV and pCMV.

3.Study population: I’m confused with the inclusion and exclusion criteria. What’s the protocol for CMV screening in the unit? Are there any changes in CMV screening during the 11-year period? When was CMV testing done for these pCMV infants?

4.Results: There is no need to repeat the data in the results in text, which have been shown in the tables very clearly. Are there any infants received Ganciclovir therapy?

5. Discussion: There is logistic problem in the first sentence of the last paragraph on Page 7. “but the 2001 NICHD standard shows significant differences in stratified diagnosis”, why “but” here?

6.The discussion on the high viral load and BPD is rather general and far fetched. More evidence should be provided to demonstrate the relationship between high CMV load and severity of BPD.

7.What are the possible infection routes for these pCMV infants?

8.In the limitations the author mentioned “the control group comprised both untested infants and those with negative test results”. I think there is a high probability of bias here, affecting the credibility of the results.

9.There are also a lot of formatting issue in the tables.

Reviewer #2: Thank you for the opportunity to review "postnatal cytomegalovirus infection and bronchopulmonary dysplasia in very low birth weight infants: influence of diagnostic criteria and viral load in a propensity score matched cohort study." The authors ask a novel study question of if postnatal CMV infection is associated with 3 clinically distinct definitions of BPD using propensity score matching and hypothesize that an association will be identified. The authors found that the association of pCMV infection with BPD was influenced by diagnostic criteria of BPD and that higher viral load was associated with more severe respiratory morbidity. These findings have biological plausibility and represent a potentially modifiable risk factor for BPD. Thank you for addressing my concerns.

Major concerns:

1) Matching was performed using GA, BW, SGA, sex, and birth year; however, even after matching, infants with pCMV had higher rates of late-onset sepsis, transfusion, steroid use, diuretic exposure, and longer respiratory support. These differences likely reflect underlying illness severity, which may simultaneously increase the probability of CMV testing and the risk of BPD. This limitation should be acknowledged more explicitly, and conclusions should avoid overstating causality. These imbalances also suggest notable residual confounding and potentially incomplete control of early illness severity. Did the authors consider using a more robust 1:2, 1:3, or 1:4 match? This more robust approach may improve the match and would strengthen the results. Alternatively, the authors could exact match on GA week, BW category/SGA status, sex, birth year, and use propensity score matching on a number of other variables to improve the causal inference of the study. In any case, the authors should provide balance diagnostics (i.e., standardized mean differences before and after matching) for the cohort to help the orient the reader to the strength of the results.

2) I think using a limited set of variables to perform the match with is justified in the sense that GA, BW, sex likely contribute the most to an infant's severity of illness and risk of BPD. However, did the authors consider whether additional early clinical factors available prior to CMV testing (i.e., early IMV, surfactant use, Apgar scores, early sepsis evaluations) should have been included?

3) Several adjusted covariates including transfusions, diuretics, glucocorticoids, and late-onset sepsis may occur after pCMV infection and therefore function as effect modifiers or mediators and not confounders. Globally, we do this all of the time in neonatology studies given the nature of the research and difficulty of parsing out time-varying covariates in our infants with outcomes at one point in time based on clinical diagnostic criteria as opposed to a pathologic diagnosis irrespective of time. However, including these variables may bias effect estimates. Did the authors consider a sensitivity analysis to only include variables that would occur before pCMV diagnosis?

Minor concerns:

1) The manuscript currently emphasizes that newer definitions “better capture” the severity attributable to pCMV. Although plausible, alternative explanations should be acknowledged such as the 2001 NICHD definition misclassifying infants on noninvasive respiratory support but low FiO2 as mild BPD. Alternatively, modern criteria classify infants based on respiratory support modality, which may track with prolonged illness trajectories regardless of pCMV etiology. Differences may reflect criteria sensitivity, not necessarily pathophysiologic differences. The discussion could be expounded upon with this nuance.

2) For the tables, please use a consistent case. For example Table 2, capitalize the first letter of the first word throughout. There are also numerous instances of variability in the authors referring to BPD severity as mild, moderate, or severe or Grade I-III (or 1-3 depending on the definition). Being consistent throughout would help the uninitiated reader.

**Do you want your identity to be public for this peer review?** For information about this choice, including consent withdrawal, please see our For information about this choice, including consent withdrawal, please see our Privacy Policy .

Reviewer #1: No

Reviewer #2: No

---

## [Author Response · Author response to Decision Letter 1]

17 Feb 2026

Response Letter

Manuscripts number: PONE-D-25-54667

Title: Postnatal cytomegalovirus infection and bronchopulmonary dysplasia in very low birth weight infants: Influence of diagnostic criteria and viral load in a propensity score-matched cohort study

Dear Editors and Reviewers,

We appreciate the opportunity to revise our manuscript titled "Postnatal cytomegalovirus infection and bronchopulmonary dysplasia in very low birth weight infants: Influence of diagnostic criteria and viral load in a propensity score-matched cohort study" and are grateful for the insightful comments provided by the reviewers. Those comments are all valuable and very helpful for revising and improving our paper, as well as the important guiding significance to our researches. In the following section, we provid detailed responses to each comments. Specifically, the comments are presented in italics, our responses are shown in red, and changes to the manuscript are marked in yellow.

Journal Requirements:

Response：We appreciate the reminder. We have meticulously examined the PLOS ONE style templates and have revised the manuscript to ensure complete adherence to all formatting and style requirements. Additionally, we have adhered to the specified file naming conventions.

“National Key Research and Development

Program of China, Grant/Award Number:

2022YFC2703500; 4+X Clinical Research

Project of Women's Hospital, School of

Medicine, Zhejiang University,

Grant/Award Number: ZDFY2021‐4X205”

Response：Thank you for the reminder. It is important to note that the funding entities did not influence the study's design, data collection and analysis, decision to publish, or the preparation of the manuscript. As per your request, this statement has been incorporated into the cover letter.

Response：We appreciate your identification of the inconsistency in the grant information. We have conducted a thorough review and made the necessary corrections to ensure alignment between the 'Funding Information' and 'Financial Disclosure' sections.

4.Please ensure that you include a title page within your main document. You should list all authors and all affiliations as per our author instructions and clearly indicate the corresponding author.

Response：We appreciate your reminder. In accordance with the journal's author guidelines, we have included a dedicated title page in the revised manuscript. This title page enumerates all authors and their full affiliations. The corresponding author is distinctly identified, with their contact information, including email and postal address, provided.

5.Please amend your manuscript to include your abstract after the title page.

Response：We appreciate your bringing this matter to our attention. In response, we have revised the manuscript to position the abstract following the title page, in accordance with the specified requirements.

6.Please include a separate caption for each figure in your manuscript.

Response：We appreciate your bringing this matter to our attention. We have revised the manuscript to include distinct descriptive captions for each figure, ensuring that each caption clearly articulates the content and significance of the corresponding figure.

7.Please include your tables as part of your main manuscript and remove the individual files. Please note that supplementary tables (should remain/ be uploaded) as separate "supporting information" files

Response：Thank you for your reminder. We've revised the manuscript to incorporate all essential tables directly into it, removing their separate files.

8.If the reviewer comments include a recommendation to cite specific previously published works, please review and evaluate these publications to determine whether they are relevant and should be cited. There is no requirement to cite these works unless the editor has indicated otherwise.

Response：We appreciate your notification regarding the editorial policy. We acknowledge receipt of this policy and affirm our commitment to complying with its stipulations. Furthermore, we confirm that, in the current review round, none of the reviewers suggested the citation of specific publications. Consequently, this policy does not pertain to the present revisions.

Reviewer #1: Thank you for the opportunity to review the manuscript entitled “Postnatal cytomegalovirus infection and bronchopulmonary dysplasia in very low birth weight infants: influence of diagnostic criteria and viral load in a propensity score-matched cohort study”. This study addresses an important and clinically relevant issue in neonatal intensive care-the association between pCMV infection and BPD in VLBWIs. However, several major methogological and interpretive concerns should be addressed before its being considered for publication.

1.The manuscript needs careful writing editing. Many sentences contain grammatical mistakes and wrong sentence structure, which make the article difficult to understand.

Response：We sincerely appreciate the reviewer's valuable feedback. We polished the language by the editing service to improve readability of the manuscript. The text has been carefully edited by a professional English language editor to ensure clarity and readability. We have also double-checked the entire manuscript to eliminate any remaining language issues.

2.Introduction: Necessary references need to be added to the concept of cCMV and pCMV.

Response：We have added concept and reference to definite congenital CMV (cCMV) and postnatal CMV

(pCMV) in the introduction section：Congenital CMV (cCMV) infection is diagnosed via polymerase chain reaction (PCR) or viral culture of urine samples, respiratory tract secretions, blood, or cerebrospinal fluid obtained within the first 3 weeks of life. Postnatal cytomegalovirus (pCMV) infection is defined as a cytomegalovirus infection after 3 weeks of age upon the exclusion of cCMV infection (P4 L65-69)

The added references :

[8] Leruez-Ville M, Chatzakis C, Lilleri D, Blazquez-Gamero D, Alarcon A, Bourgon N, et al. Consensus recommendation for prenatal, neonatal and postnatal management of congenital cytomegalovirus infection from the European congenital infection initiative (ECCI). The Lancet Regional Health – Europe. 2024;40. doi:10.1016/j.lanepe.2024.100892

3.Study population: I’m confused with the inclusion and exclusion criteria. What’s the protocol for CMV screening in the unit? Are there any changes in CMV screening during the 11-year period? When was CMV testing done for these pCMV infants?

Response : We appreciate the reviewer's concern regarding the CMV screening protocol. In our study, the CMV screening protocol remained consistent throughout the 11-year period and added in the manuscript：

To rule out cCMV, all preterm infants were routinely screened via urine and respiratory secretion samples within the first 48–72 h and repeated 2–3 weeks after birth. A pCMV infection was diagnosed when testing was prompted by clinical suspicion beyond 3 weeks of age based on relevant symptoms (e.g., sepsis-like syndrome, gastrointestinal manifestations) or laboratory abnormalities (e.g., thrombocytopenia, transaminitis).(P6 L94-98)

4.Results: There is no need to repeat the data in the results in text, which have been shown in the tables very clearly. Are there any infants received Ganciclovir therapy?

Response : Thank you for the suggestion. We've streamlined the Results section by removing repetitive descriptions of numerical data already shown in the tables. The text now emphasizes key trends, highlights significant findings, and offers interpretation, directing readers to the tables for specific values.

In our cohort, 28/74 (37.8%) infants with postnatal cytomegalovirus (pCMV) infection were administered ganciclovir therapy. To account for this variable, we incorporated "ganciclovir treatment (yes/no)" as a covariate in the fully adjusted Model 5 of our regression analysis. This adjustment has been explicitly described in the Statistical Analysis (P7 L129 )and Results sections(P14 L171) of the revised manuscript.

5. Discussion: There is logistic problem in the first sentence of the last paragraph on Page 7. “but the 2001 NICHD standard shows significant differences in stratified diagnosis”, why “but” here?

Response : Thank you for pointing out the logical inconsistency. We have revised the sentence to clarify the intended contrast. The revised sentence now reads: Our study found significant differences in the 2018 NICHD and 2019 NRN criteria between pCMV-infected and non-pCMV-infected groups in the overall incidence and Grades II–III/2–3. In contrast, the 2001 NICHD criteria revealed significant differences between the moderate and severe groups only. However, the overall diagnostic rate did not differ significantly between groups.(P21 L225-228)

6.The discussion on the high viral load and BPD is rather general and far fetched. More evidence should be provided to demonstrate the relationship between high CMV load and severity of BPD.

Response :We appreciate the reviewer's constructive feedback. In the revised manuscript, we have strengthened the discussion by including additional references and data，these additions can be found in the discussion sections.

Our findings indicate that an elevated CMV viral load (≥1.34 × 10⁵ copies/mL) is associated with an increased incidence of BPD as well as worsened respiratory outcomes, prolonged IMV duration, and extended respiratory support in VLBW infants (Table 5). This aligns with prior reported that a high CMV DNA load (>50,000 copies/mL) was independently associated with organ hemorrhage (OR, 3.61; 95% CI, 1.03–12.65) [37]. Additionally, a higher viral load has been correlated with more severe manifestations and poorer outcomes in cases of CMV pneumonia among immunocompetent hosts and CMV disease in transplant recipients [38, 39]. Although specific thresholds have not been established, a higher viral load is associated with symptomatic infection and poor prognosis [40–42]. The potential mechanisms by which a high CMV load exacerbates BPD are multifaceted. First, elevated viremia can trigger a systemic inflammatory response or viral sepsis characterized by the release of proinflammatory cytokines [43]. This systemic inflammation may exacerbate the inflammatory environment in the developing preterm lung, thereby disrupting alveolarization and promoting fibrosis, the key features of BPD pathogenesis. Second, the CMV induces endothelial injury and dysfunction [44]. Within the pulmonary vasculature, this can impair angiogenesis and vascular remodeling, further worsening BPD severity. Finally, human CMV (HCMV) infection is typically managed by a robust immune response; however, in individuals with compromised immune systems, HCMV can replicate at high levels, leading to end-organ disease (EOD) [45]. In compromised hosts, particularly neonates who are preterm or have a VLBW, the absence of T cell and natural killer (NK) cell function facilitates unrestricted viral replication. This results in high viral loads and ultimately culminates in EOD, which is characterized by direct viral damage, such as retinitis and pneumonia, and indirect effects, including endothelial injury [46,47]. However, prior research predominantly concentrated on cCMV infections; thus, the relationship between viral load and disease severity in pCMV infections requires further exploration. (P21-22 L244-265)

7.What are the possible infection routes for these pCMV infants?

Response : Thank you for your question, it is posited that the primary mode of transmission is through exposure to breast milk. Due to the long time span, much data of breast milk was missing and not included in the discussion. We addde the possible infection routes in the passage: Possible routes of pCMV infection in infants include breast milk, blood products, and horizontal transmission. In the current era, with the reduction in CMV transmission via blood products achieved through seronegative donor selection and leukoreduction, exposure to breast milk from CMV-seropositive mothers has emerged as the most common transmission route.(P4-5 L69-72)

8.In the limitations the author mentioned “the control group comprised both untested infants and those with negative test results”. I think there is a high probability of bias here, affecting the credibility of the results.

Response :

We appreciate the reviewer's concern regarding potential bias in the control group composition. We acknowledge this as a limitation in the study design. The absence of universal CMV screening could lead to misclassification bias, potentially resulting in the omission of asymptomatic pCMV infections within the control group. Importantly, such non-differential misclassification is expected to bias the findings towards the null hypothesis, thereby underestimating the true effect size. Consequently, the significant associations we have reported likely represent conservative estimates. We will address and clarify this point in the revised Limitations section of the manuscript: Although this method aligns with real-world clinical practice and is prevalent in the literature [23,51], it introduces a potential misclassification bias. Therefore, some asymptomatic pCMV infections may have been overlooked in the control group. This non-differential misclassification is likely to bias our findings toward the null hypothesis by attenuating the intergroup differences. Consequently, the observed associations between pCMV infection and BPD, particularly when newer diagnostic criteria were applied, are likely conservative estimates of the true effect. (P23 L274-279)

9.There are also a lot of formatting issue in the tables.

Response :We have carefully reviewed all tables in the manuscript and corrected the formatting issues. The tables have been standardized to ensure consistent font size, alignment, and spacing throughout the document. Additionally, we have ensured that all table headers are clearly labeled and that the data is presented in a clear and organized manner.

Reviewer #2: Thank you for the opportunity to review "postnatal cytomegalovirus infection and bronchopulmonary dysplasia in very low birth weight infants: influence of diagnostic criteria and viral load in a propensity score matched cohort study." The authors ask a novel study question of if postnatal CMV infection is associated with 3 clinically distinct definitions of BPD using propensity score matching and hypothesize that an association will be identified. The authors found that the association of pCMV infection with BPD was influenced by diagnostic criteria of BPD and that higher viral load was associated with more severe respiratory morbidity. These findings have biological plausibility and represent a potentially modifiable risk factor for BPD. Thank you for addressing my concerns.

Major concerns:

1) Matching was performed using GA, BW, SGA, sex, and birth year; however, even after matching, infants with pCMV had higher rates of late-o

---

## [Decision Letter · Decision Letter 1]

16 Mar 2026

Dear Dr. Zhu,

Thank you for submitting your manuscript to PLOS ONE. After careful consideration, we feel that it has merit but does not fully meet PLOS ONE’s publication criteria as it currently stands. Therefore, we invite you to submit a revised version of the manuscript that addresses the points raised during the review process.

We look forward to receiving your revised manuscript.

Kind regards,

Kazumichi Fujioka

Academic Editor

PLOS One

Journal Requirements:

Reviewers' comments:

Reviewer's Responses to Questions

**Comments to the Author**

Reviewer #2: All comments have been addressed

2. Is the manuscript technically sound, and do the data support the conclusions?

Reviewer #2: Partly

3. Has the statistical analysis been performed appropriately and rigorously?

Reviewer #2: Yes

4. Have the authors made all data underlying the findings in their manuscript fully available?

Reviewer #2: Yes

5. Is the manuscript presented in an intelligible fashion and written in standard English?

Reviewer #2: Yes

Reviewer #2: The revised manuscript is improved compared with the prior version. The authors have clarified several aspects of the analytic approach, particularly the matching strategy, and the inclusion of standardized mean differences (SMDs) before and after matching meaningfully improves the transparency of the analysis. The additional modeling framework incorporating pre-diagnosis clinical variables is also helpful and provides useful context for the primary findings. Overall, I believe the authors have made thoughtful revisions in response to the initial review.

However, I continue to have some concern regarding the specification of the matching strategy and the residual imbalance that remains in one of the matching variables. The current analysis matches on gestational age, birth weight, SGA status, sex, and birth year. Because SGA is intrinsically derived from the relationship between gestational age and birth weight, including all three variables simultaneously in the matching algorithm may introduce redundancy and may make it more difficult to achieve optimal covariate balance, particularly given the relatively small exposed cohort. Consistent with this concern, the post-matching balance diagnostics indicate that SGA remains the most imbalanced variable after matching, with a standardized mean difference exceeding 0.20, whereas gestational age and birth weight appear reasonably well balanced. Given that SGA was itself included as a matching variable, this residual imbalance is notable and raises the possibility that the current matching specification may be somewhat over-constrained.

The authors may wish to consider whether a different matching approach would provide improved balance and interpretability. For example, matching on gestational age, birth weight, sex, and birth year alone would capture the primary baseline determinants of prematurity-related risk while avoiding potential redundancy introduced by including SGA simultaneously with gestational age and birth weight. SGA could then be incorporated as a covariate in adjusted models rather than as a matching variable. Alternatively, if the authors prefer to retain the current matching specification, the residual imbalance in SGA should be explicitly acknowledged in the manuscript. Specifically, the manuscript should note that SGA remained imbalanced after matching (SMD >0.20), discuss the potential implications for interpretation of the findings, and clarify that this may reflect the close relationship between gestational age, birth weight, and SGA status. Addressing this issue, either by reconsidering the matching specification or by more explicitly acknowledging the remaining imbalance and its implications, would strengthen the methodological clarity of the manuscript.

**Do you want your identity to be public for this peer review?** For information about this choice, including consent withdrawal, please see our For information about this choice, including consent withdrawal, please see our Privacy Policy .

Reviewer #2: No

---

## [Author Response · Author response to Decision Letter 2]

23 Mar 2026

Dear Editors and Reviewers,

We would like to express our sincere gratitude for your thoughtful and constructive comments on our manuscript. These comments are highly valuable and instrumental for revising and improving our paper, and offer significant guidance for our research. In the following section, we provide detailed responses to the comments. Specifically, the comments are presented in italics, our responses are shown in red, and changes to the manuscript are marked in yellow.

Reviewer #2: ...The authors may wish to consider whether a different matching approach would provide improved balance and interpretability. For example, matching on gestational age, birth weight, sex, and birth year alone would capture the primary baseline determinants of prematurity-related risk while avoiding potential redundancy introduced by including SGA simultaneously with gestational age and birth weight. SGA could then be incorporated as a covariate in adjusted models rather than as a matching variable. Alternatively, if the authors prefer to retain the current matching specification, the residual imbalance in SGA should be explicitly acknowledged in the manuscript. Specifically, the manuscript should note that SGA remained imbalanced after matching (SMD >0.20), discuss the potential implications for interpretation of the findings, and clarify that this may reflect the close relationship between gestational age, birth weight, and SGA status. Addressing this issue, either by reconsidering the matching specification or by more explicitly acknowledging the remaining imbalance and its implications, would strengthen the methodological clarity of the manuscript.

Response：Thank you for this valuable comment. The insights regarding our propensity score matching strategy are highly valued and have helped us improve the clarity and rigor of our work. We agree that the post-matching standardized mean difference for SGA (SMD > 0.20) indicates a residual imbalance, despite adequate balance on GA and BW. Following your constructive suggestion, we have explicitly acknowledged this limitation. Accordingly, we have made the following revisions to the manuscript:

In the Results section (following Table 2), we have now explicitly noted the residual imbalance in SGA after matching: The post-matching diagnostic assessments indicated that the majority of variables were well-balanced. However, a residual imbalance in SGA status remained (SMD = 0.228). (P8 L140-142)

In the Discussion section (Limitations), we have added discuss to this issue: Second, an imbalance persisted in the SGA following the matching process (SMD = 0.228), potentially attributable to the somewhat over-constrained simultaneous matching for SGA, GA, and BW, in conjunction with the relatively small size of the exposed cohort (n = 74). Although this imbalance was accounted for in the regression models, residual confounding may still be present. Nevertheless, the effect estimates across various models were consistent, indicating that the impact of such confounding is likely minimal. (P23 L278-283)

We believe that these revisions adequately address the concerns you raised and enhance the clarity and transparency of our study.

---

## [Editor Report · Decision Letter 2]

25 Mar 2026

Postnatal cytomegalovirus infection and bronchopulmonary dysplasia in very low birth weight infants: Influence of diagnostic criteria and viral load in a propensity score–matched cohort study

PONE-D-25-54667R2

Dear Dr. Zhu,

We’re pleased to inform you that your manuscript has been judged scientifically suitable for publication and will be formally accepted for publication once it meets all outstanding technical requirements.

Kind regards,

Kazumichi Fujioka

Academic Editor

PLOS One

Additional Editor Comments (optional):

Well Revised
---

## [Editor Report · Acceptance letter]

PONE-D-25-54667R2

PLOS One

Dear Dr. Zhu,

I'm pleased to inform you that your manuscript has been deemed suitable for publication in PLOS One. Congratulations! Your manuscript is now being handed over to our production team.

Kind regards,

on behalf of

Dr. Kazumichi Fujioka

Academic Editor

PLOS One